# Potential Molecular Biomarkers of Central Nervous System Damage in Breast Cancer Survivors

**DOI:** 10.3390/jcm11051215

**Published:** 2022-02-24

**Authors:** Maria Pospelova, Varvara Krasnikova, Olga Fionik, Tatyana Alekseeva, Konstantin Samochernykh, Nataliya Ivanova, Nikita Trofimov, Tatyana Vavilova, Elena Vasilieva, Maria Topuzova, Alexandra Chaykovskaya, Albina Makhanova, Anna Mikhalicheva, Tatyana Bukkieva, Kenneth Restor, Stephanie Combs, Maxim Shevtsov

**Affiliations:** 1Personalized Medicine Centre, Almazov National Medical Research Centre, 197341 Saint Petersburg, Russia; pospelovaml@mail.ru (M.P.); varya.krasnikova.93@mail.ru (V.K.); fvolga@mail.ru (O.F.); atmspb@mail.ru (T.A.); neurobaby12@gmail.com (K.S.); ivamel@yandex.ru (N.I.); nikita.trofimov.1999@mail.ru (N.T.); vtv.lab.spb@gmail.com (T.V.); elena-almazlab@yandex.ru (E.V.); topuzova_mp@almazovcentre.ru (M.T.); mumu6394@gmail.com (A.C.); a.mahanova.a@mail.ru (A.M.); kinefanna@yandex.ru (A.M.); tanya-book25@mail.ru (T.B.); 2Nursing Programme, University of St. Francis, Joliet, IL 60435, USA; kennethdmrestor@gmail.com; 3Department of Radiation Oncology, Klinikum rechts der Isar, Technishe Universität München (TUM), 81675 Munich, Germany; stephanie.combs@tum.de; 4National Center for Neurosurgery, Nur-Sultan 010000, Kazakhstan

**Keywords:** breast cancer, adhesion molecules, neuron-specific enolase, antibodies to NMDA receptors, PECAM-1, ICAM-1, central nervous system damage

## Abstract

Damage of the central nervous system (CNS), manifested by cognitive impairment, occurs in 80% of women with breast cancer (BC) as a complication of surgical treatment and radiochemotherapy. In this study, the levels of ICAM-1, PECAM-1, NSE, and anti-NR-2 antibodies which are associated with the damage of the CNS and the endothelium were measured in the blood by ELISA as potential biomarkers that might reflect pathogenetic mechanisms in these patients. A total of 102 patients enrolled in this single-center trial were divided into four groups: (1) 26 patients after breast cancer treatment, (2) 21 patients with chronic brain ischemia (CBI) and asymptomatic carotid stenosis (ICA stenosis) (CBI + ICA stenosis), (3) 35 patients with CBI but without asymptomatic carotid stenosis, and (4) 20 healthy female volunteers (control group). Intergroup analysis demonstrated that in the group of patients following BC treatment there was a significant increase of ICAM-1 (mean difference: −368.56, 95% CI −450.30 to −286.69, *p* < 0.001) and PECAM-1 (mean difference: −47.75, 95% CI −68.73 to −26.77, *p* < 0.001) molecules, as compared to the group of healthy volunteers. Additionally, a decrease of anti-NR-2 antibodies (mean difference: 0.89, 95% CI 0.41 to 1.48, *p* < 0.001) was detected. The intergroup comparison revealed comparable levels of ICAM-1 (mean difference: −33.58, 95% CI −58.10 to 125.26, *p* = 0.76), PECAM-1 (mean difference: −5.03, 95% CI −29.93 to 19.87, *p* = 0.95), as well as anti-NR-2 antibodies (mean difference: −0.05, 95% CI −0.26 to 0.16, *p* = 0.93) in patients after BC treatment and in patients with CBI *+* ICA stenosis. The NSE level in the group CBI *+* ICA stenosis was significantly higher than in women following BC treatment (mean difference: −43.64, 95% CI 3.31 to −83.99, *p* = 0.03). Comparable levels of ICAM-1 were also detected in patients after BC treatment and in the group of CBI (mean difference: −21.28, 95% CI −111.03 to 68.48, *p* = 0.92). The level of PECAM-1 molecules in patients after BC treatment was also comparable to group of CBI (mean difference: −13.68, 95% CI −35.51 to 8.15, *p* = 0.35). In conclusion, among other mechanisms, endothelial dysfunction might play a role in the damage of the CNS in breast cancer survivors.

## 1. Introduction

Every year, at least 1.6 million patients are newly diagnosed with breast cancer (BC) worldwide and the incidence is growing [1]. Breast cancer mortality is second after lung cancer, with 70% of deaths occurring in low-income countries [2]. Modern screening programs and the development of combined treatment regimens contributed to a reduction in mortality rates of approximately 30% between 1991 and 2015 [3] and in high-income countries, the 5-year survival rate of patients with BC reaches a value of 91.1% [4]. However, radical treatment regimens result in an increase in chronic treatment-related complications, including upper limb lymphedema, persistent pain syndrome, shoulder joint contractures, musculoskeletal disorders, peripheral polyneuropathy, and psychoemotional disorders [5]. In recent years, disorders of the central nervous system (CNS) have been studied in more detail.

Already by 1980, a reduction in cognitive functions could be associated with adjuvant chemotherapy in women with BC [6]. According to current data, at least 80% of patients after surgical treatment combined with radiochemotherapy developed cognitive impairment such as reduced speed of information processing, attention, memory, and executive function [7]. These disorders persist for many years after treatment, [8] which negatively impact the patient’s quality of life, [9] and their ability to work [10]. It is assumed that not only does adjuvant chemotherapy impairs cognition but also surgery, radiation, and anti-estrogenic therapy can impair brain structure and functioning [11,12,13]. Brain damage in treated women with breast malignancies could be confirmed by neuroimaging studies. Changes in the concentration of metabolites in the CNS and the integrity of the white matter were detected by magnetic resonance spectroscopy and diffusion tensor imaging in breast cancer survivors [14]. According to multimodal magnetic resonance imaging (MRI) data, patients have significant changes in the brain that were not present before chemotherapy [15]. These findings include a decrease in the density of the gray matter of the brain, and structural change of the white matter, especially in the frontal lobes and hippocampus, which are persistent after treatment [16,17]. Even 5–10 years following treatment, specific changes in the frontal cortex, cerebellum, and basal ganglia are detectable by positron emission tomography (PET) [18].

The pathogenic mechanisms of CNS damage in BC patients after radical chemotherapy are still not completely understood. Previously, it was assumed that adjuvant chemotherapy exerts a direct toxic effect on oligodendrocytes and CNS progenitor cells that cause brain damage, and therefore the term “Chemical Brain” or “Chemobrain” was created [19]. However, later changes in the integrity of the white matter were also found after other treatment regimens including radiotherapy, chemotherapy and, therefore, the term was changed to “cancer-related cognitive impairment” [20,21]. Several theories exist to try to explain the mechanisms of the treatment-related brain damage, but the etiology of the CNS damages is most likely multifactorial. A direct neurotoxic effect of chemotherapeutic agents such as cisplatin, paclitaxel, BCNU (1,3-bis (2-chloroethyl)-1-nitrosourea) to the CNS has been investigated and most of these chemotherapeutic agents are able to penetrate the blood–brain barrier (BBB) [22]. Furthermore, deoxyribonucleic acid (DNA) damages can also cause neurodegeneration after chemotherapy [23]. The disruption of the DNA structure by oxidative stress in CNS cells can result in misfolded proteins and apoptosis of neurons [24]. Given that radiation can cause breaks in DNA strands, the combined treatment of radiation and chemotherapy results in even greater damaged to the CNS cells. Moreover, some research also suggested that an imbalance of cytokines can induce brain damage [25] i.e., the number of proinflammatory cytokines like interleukin-6 (IL-6), interleukin-8 (IL-8), and tumor necrosis factors (TNC) are significantly increased after therapy for malignant tumors [26]. Some studies suggested that chemotherapy and radiation therapy accelerate biological aging including that of the brain, which causes a change in the neuropsychological status that is detected by imaging methods [27]. Research has also shown that 20–30% of patients with BC have decreased cognitive abilities even before the start of treatment [28]. A tumor-mediated increase in pro-inflammatory cytokines with neurotoxic properties is assumed to contribute to a premature neurodegenerative process i.e., ineffective DNA repair mechanisms [22].

Other CNS characteristics found from those post-treatment symptoms in BC survivors include cerebrovascular disorders in the vertebrobasilar basin after surgery. These complications are induced from the spasm of the anterior scalenus muscles in the thoracic outlet syndrome on the neurovascular bundle after surgery, chemotherapy, and/or radiation therapy. Consequently, these results in the development of vertebral-basilar insufficiency and cognitive-emotional disorders in more than 70% of the patients undergoing treatment. These results in cognitive impairment are shown as a decrease in verbal memory and attention, static-locomotor, and dynamic ataxia [29].

Since the post-treatment pathogenesis of CNS damage in BC patients is not yet fully understood i.e., direct toxic effect, ischemia, hypoxia, endothelial dysfunction, apoptosis, cellular degeneration, to name a few, it remains difficult to conduct primary prevention against these cellular damages. However, the characterization of molecular mechanisms underlying brain damage may provide new opportunities for neuroprotection during treatment and follow-up periods.

In this study, various biomarkers were measured using the patient’s blood that might show evidence for CNS cellular damage. Elevated levels of the intercellular adhesion molecule type-1 (ICAM-1), endothelial and platelet adhesion molecule type-1 (PECAM-1), neuron-specific enolase (NSE), and antibodies recognizing the NR-2 subunit of N-methyl-D-aspartate receptor (NMDA) receptors (anti-NR2 antibodies) have been found in breast cancer survivors. The properties of these biomarkers will be discussed below.

ICAM-1 is a cell membrane glycoprotein that is expressed on the immune, epithelial, and endothelial cells. As a marker for inflammation, ICAM-1 is increased in atherosclerosis [30], chronic obstructive pulmonary disease (COPD) [31], sepsis [32], and age-related cognitive disorders [33]. ICAM-1 also plays an important role in endothelial cells since it regulates the permeability of the vascular wall in response to inflammatory processes [34]. In some studies, ICAM-1 is also considered a marker for endothelial dysfunction [35]. PECAM-1 is a glycoprotein expressed in the endothelial and several hematopoietic cells. The main function of PECAM-1 is the regulation of vascular wall permeability and the immune system [36]. There is an increased concentration of soluble PECAM-1 in disorders like atherosclerosis [37], neurodegenerative diseases [38], rheumatoid arthritis (RA), and metabolic syndrome [39].

These studies show that ICAM-1 and PECAM-1 are considered to be highly sensitive markers for endothelial dysfunction [40]. In relation to the pathogenesis of the CNS lesions in post-treatment BC patients, several researchers have identified endothelial damage to these areas. The biomarkers like ICAM-1 and PECAM-1 can be utilized in the detection of these endothelial damages [41] because increased levels of these biomarkers are indicative of chronic inflammation and endothelial dysfunction, which are also the characteristics of the CNS in post-treatment BC patients.

The NSE is a dimeric isoenzyme of the glycolytic enolase enzyme and is localized mainly in neurons and cells of neuroendocrine origin [42]. Elevated levels of NSE are considered as a marker for oxidative stress in neurons, as well as abnormal activation of the astrocytes and microglia [43]. When BBB becomes dysfunctional and neuronal cell damage occurs, there is a significant increase of NSE levels in the spinal fluid prior to NSE blood levels [44]. To date, the determination of the serum level of this enzyme is used for diagnosis and treatment monitoring of many tumors including neuroblastomas, non-small cell and small cell lung cancer, pheochromocytomas, and neuroendocrine tumors [45].

On one hand, there are also studies proving the diagnostic value of NSE in detecting intracranial BC metastases [46]. In addition, NSE values can also be used as a marker to quantify the extent of brain injuries like ischemic stroke, intracerebral hemorrhage, and cerebral hypoxemia after cardiopulmonary resuscitation [47]. On the other hand, some studies detected increased levels of NSE in psychiatric disorders like bipolar disorder and depression [48]. In the current research, the NSE was considered to be the biomarker used in detecting the local and systemic effects of BC treatments in the CNS [49].

In women after BC treatment, the NSE can be used to determine oxidative damages to the neurons and the pathological activation of microglia, which are considered the leading mechanisms for cerebral disorders after treatment of BC.

The anti-NR-2 antibodies are specific immunoglobulins that are produced when components from destroyed NMDA receptors enter the bloodstream [50]. The NMDA receptors are mainly present in the neurons where they regulate synaptic activity [51]. During an acute or chronic cerebral ischemia, a cascade of pathological complexes is triggered, which makes excitatory molecules such as glutamate enter into the extracellular space that in turn results in neuronal apoptosis due to their neurotoxic properties [52].

Elevated levels of anti-NR-2 antibodies are also detected after ischemic and transient ischemic attacks (TIA) [53], as well as in other diseases affecting the CNS (i.e., systemic lupus erythematosus (SLE), paraneoplastic encephalitis, epilepsy, mania) [54]. In relation to the insufficient blood supply to the brain, chronic ischemia in the vertebrobasilar basin is considered one of the main causes of damage to the CNS after BC treatments [55]. For this reason, the authors chose anti-NR-2 antibodies as the biomarker for detecting ischemic cells in BC survivors and to explore the neurotoxic properties of glutamate secondary to chronic ischemia in the CNS.

In the current study, a comparative assessment of CNS and endothelial lesions biomarkers (i.e., ICAM-1, PECAM-1, NSE, and anti-NR-2 antibodies), was performed in BC patients with post-treatment symptoms and in both with chronic ischemic brain disease with and without hemodynamic asymptomatic carotid stenosis. The levels of the molecules were compared to the 20 age-matched female healthy volunteers.

Patients with confirmed chronic ischemic brain damage were chosen as the comparison group to thoroughly explore the extent of CNS damage in BC patients in the post-treatment follow-up period. By comparing the results of these studies in elderly patients with marked changes in the CNS and in young women after treatment of BC, this research study can help to explain the extent of both structural and functional abnormalities in the CNS caused by antitumor treatment.

In this research, the men were excluded from the study due to the low prevalence of BC diagnosis—less than 1% of all BC cases [56]—which can distort the statistical processing of data including the differences of both sexes in the expression of intercellular adhesion molecules and the production of proinflammatory cytokines. The researchers also found no information about the biomarker concentration in men with BC. These are some of the reasons why women with BC in post-treatment were chosen for this study [57]. Since the main purpose of this study is to examine a group of BC patients with post-treatment symptoms, female healthy volunteers were also included in the control group in order to obtain more objective results.

## 2. Materials and Methods

### 2.1. Experimental Design

The study was carried out in compliance with the principles of the Helsinki Declaration of the World Medical Association with the consent from the Ethics Committee of the Federal State Budgetary Institution, “Almazov National Medical Research Center”, of the Ministry of Health of the Russian Federation (conclusion of 31.10.2019; protocol number 10). All patients in the study and healthy volunteers signed an informed consent. There were no identifiable potential risks to the patients during this study. An open single-center controlled study of biomarkers of CNS and endothelial lesions-ICAM-1, PECAM-1, NSE, and anti-NR-2 antibodies, was conducted. The study had a total of 102 subjects that were divided into four groups-(1) women after BC treatment (experimental group, n = 26), (2) patients with chronic cerebral ischemia (CBI) with asymptomatic internal carotid artery (ICA) stenosis (comparison group 1, CBI + ICA stenosis, n = 21), (3) patients with CBI without asymptomatic ICA stenosis (comparison group 2, n = 35), and (4) healthy female volunteers (control group, n = 20).

#### 2.1.1. Inclusion Criteria

The study included women aged 25 to 50 after BC treatments including a combination of surgical treatment (unilateral or bilateral Peutie mastectomy) and radiation therapy, a combination of surgical treatment and chemotherapy, or a complex treatment (combination of surgery, radiation therapy, and chemotherapy) [58], who developed post-treatment symptoms (including cognitive impairments, musculoskeletal symptoms, lymphedema, post-mastectomy pain syndrome (PMPS), etc.) [59] secondary to any BC treatments mentioned above. Patients without post-treatment symptoms were not included in the study.

Women with post-treatment symptoms were initially selected depending on the fact of surgery (modified Patey radical mastectomy) or surgery with external beam RT and they had musculoskeletal problems such as protective positions due to fear of movement. The patients avoid using the limb, which leads to shortening of the muscles and compression of the joint capsule and pectoral muscles. In turn, a complex of biomechanical disorders as well as long thoracic nerve damage leads to scapular winging. The women also presented with such clinical manifestations as subacromial impingement syndrome, adhesive capsulitis, symptomatic rotator cuff tendinopathy. They presented with sensory disturbances in the upper extremity and anterior chest wall, axilla, and/or shoulder. Participants had to undergo unilateral or bilateral breast surgery (with a histological verification) and, in general, represented a good health status (Eastern Cooperative Oncology Group (ECOG) performance status 0-1). They did not report cardiac, endocrine, rheumatic neuromuscular or musculoskeletal disorders and other oncological diseases.

Group 2 (comparison group 1, CBI + ICA stenosis) enrolled 21 patients, aged 54–87 years, suffering from CBI with ICA atherosclerotic stenosis (more than 70%), not showing focal neurological symptoms (asymptomatic stenosis), with compensated concomitant somatic pathology. The group included 8 men and 13 women, with an average age of 73 ± 9.5 years. The degree of stenosis was determined by the method of ultrasound triplex angioscanning on the device Vivid E95, General Electric.

Group 3 (comparison group 2, CBI) enrolled 35 patients, aged 55 to 81 years, suffering from CBI without hemodynamically significant ICA stenosis, with compensated concomitant somatic pathology. The group included 7 men and 28 women, with an average age of 67.5 ± 6.8 years. The degree of stenosis or its absence was determined by the method of ultrasound triplex angioscanning on the device Vivid E95, General Electric.

Group 4 (control group) of healthy volunteers included 20 women aged 25 to 50 years, with no history of cancer, severe somatic diseases. The average age of healthy female volunteers was 39.0 ± 5.4 years (from 27 to 42 years).

#### 2.1.2. Exclusion Criteria

Exclusion criteria in group 1 included signs of progression of the main oncological disease; the presence of distant metastases of breast cancer, including with nervous system damage, the presence of protrusions and/or hernias of the intervertebral discs of the spine, ankylosing spondylitis, pathological fractures of the vertebral bodies, acute spinal injuries, and conditions after spinal surgery; the presence of hemodynamically significant atherosclerotic stenoses of the head and neck main arteries; acute infectious and mental diseases, as well as other conditions that prevent neurological examination and manual diagnosis; pregnancy; decompensated somatic pathology; and contraindications to MRI.

Exclusion criteria in groups 2 and 3 included contraindications to MRI; acute cerebrovascular accident, demyelinating diseases, craniocerebral trauma in the anamnesis, developmental abnormalities of the brain, brain tumors; chronic somatic diseases in the decompensation stage; psychiatric diseases, hearing loss, congenital and acquired heart defects and large vessels in the decompensation stage, and thromboembolism in the pulmonary artery system in the anamnesis.

### 2.2. Clinical and Neuropsychological Assessment

Clinical, neurological (assessment of complaints, anamnesis, neurological examination), and laboratory evaluation were performed in all studied groups.

During the initial examination, the patients in experimental group (women after BC treatment) complained of headaches and dizziness, swelling of the upper limb, numbness and pain of moderate and pronounced intensity in the arm, in the scapula and in the shoulder joint on the side of the operation, restriction of movement in the upper limb.

The anamnesis included: the period after the operation, the type of operation, the course of chemotherapy, the course of radiation therapy, the presence of relapses, and the use of Tamoxifen.

Objective examination of the experimental group (women after BC treatment) included: neurological examination, measurement of the volume of the upper extremities, joint movements.

During the neurological examination, the assessment of muscle strength and surface sensitivity were studied, coordination tests were performed (finger-nasal test, Romberg test), and symptoms of damage to the brachial plexus trunks were detected (hypesthesia, paresthesia, muscle hypotension, paresis). Neurological examination was performed according to the international standard protocol [60].

Examinations included assessment of sensory perception (0—hypesthesia, 1—normal perception, 2—hyperesthesia) in areas of axilla and lower brachial plexus innervation. Test results were blinded for participants during the test procedure. Tests were first performed on the contralateral forearm and chest wall, next in the postmastectomy area and lower brachial plexus on the same side. The control group was assessed in both sides of the chest. 

Examination included an assessment of motor functions, which began with the detection of the presence of hypotrophy of the muscles of the upper shoulder girdle on the side of the operation (0—normal, 1—hypotrophy). Muscle tone was assessed (0—normotonia, 1—hypotonia). The study of the muscle strength of the upper extremities was carried out alternately on both sides, while the strength of the muscles on the contralateral side was compared (0—no movement disorder, 1—paresis). 

The study included an assessment of coordination movements. The Romberg test was used to detect statistical ataxia (0—yes, 1—no). A finger-nose test was performed to detect dynamic ataxia (0—yes, 1—no). Vestibulo-ataxic syndrome was verified by detection of static or dynamic ataxia.

The assessment of active movements in the joints of the upper extremities on both sides was carried out (0—there are no restrictions, 1—there is a restriction). 

The upper extremities were measured on both sides to assess the volume of the limb and subsequently to assess the degree of edema. Classification based on determining the circumference of an edematous limb when comparing it to a healthy limb determines four degrees of edema: I—an increase in the circumference of the affected limb to 1–2 cm; II—from 2 to 6 cm; III—from 6 to 10 cm; IV—more than 10 cm.

### 2.3. Laboratory Examination

The serum was collected from 7 mL of blood and stored at −70 °C. Determination of soluble endothelial platelet adhesion molecules 1 (sPECAM-1) was performed employing the Human sPECAM-1 ELISA according to the manufacturer’s protocol (Bender MedSystems GmbH). Assessment of intercellular adhesion molecules 1 (sICAM-1) was performed by a set of rHuman sICAM-1 kit (ELISA Bender MedSystems GmbH). The levels of neuron-specific enolase (NSE) were assessed employing Cobas E411Roch electrochemiluminescent immunoassay. Antibodies to NMDA receptors (anti-NR-2 antibodies) studied with a set of reagents for the quantitative determination of antibodies to the NR2 subunit of the NMDA glutamate receptor by immunoenzyme analysis. The microplate spectrophotometer Bio-RaD Model 680 Microplate Reader with Zemfira software was used. The optical density was measured at wavelengths of 450/655 nm. Concentrations were calculated automatically from standard curves.

### 2.4. Statistical Analysis 

Statistical processing of the obtained data was carried out using the IBM SPSS Statistics 28.0.1.0 program (IBM, Armonk, New York, NY, USA). 

All available data were analyzed statistically. To assess the qualitative variables, absolute and relative indicators (% of the number of observations) were used. Quantitative variables were characterized by means, standard deviations, medians, and ranges of values. Statistical comparison of changes in quantitative indicators of efficacy and safety relative to baseline parameters was carried out using nonparametric methods. The statistical significance of changes in quantitative indicators was checked using the Kruskal–Wallis test. Statistical comparisons of mean changes for quantitative continuous efficacy variables between four parallel groups (gr.) Were performed using the Games–Howell test. The probability of a Type I error (two-sided significance level) is set at 5%.

## 3. Results

### Clinical and Neuropsychological Evaluation of Patients

Group 1 constituted 26 patients after BC treatment. The average age of the patients was 47.0 ± 3.7 years (from 32 to 50 years). Treatment of patients with BC was finished between 1 and 10 years (4.3 ± 2.1 years). All patients underwent a Patey mastectomy (unilateral n = 22, bilateral n = 4), breast reconstruction was performed in 5 patients. Complex treatment (surgical, radiation therapy, chemotherapy) of breast cancer was performed in 16 women, a combination of surgical treatment and chemotherapy was performed in 8 patients, a combination of surgical treatment and radiation therapy—in 5 patients. Stage I (T1N0M0) was detected in 3 patients, stage II A (T2N1M0)—in 6, stage II B (T3N1M0)—in 7 patients, stage III A (T3N2M0)—5, stage III B (T4N2M0)—5 (Table 1).

Nervous system complaints were observed in all women after breast cancer treatment. The most common symptoms included headache (n = 13, 50%), dizziness (n = 15, 57%), memory degradation (n = 9, 34%), fatigue (n = 11, 42%), and sleep disturbances (n = 5, 20%). All patients presented with clinical neurological symptoms after treatment: in addition to edema of the upper limb on the side of surgical treatment (n = 11, 42%), there were: decreased sensitivity of the upper limb (n = 16, 61%), paresthesia or hypoesthesia (n = 8, 30%), muscle weakness (n = 18, 69%), limited movement in the shoulder joint (n = 8, 30%), pain in the upper limb (n = 13, 50%) and the upper arm (n = 11, 42%), neurological examination revealed vestibulo-atactic syndrome in 12 patients (46%), clinical manifestations of polyneuropathy—in 11 patients (42%). (%). ECOG 0 status was in 8 (31%) patients and ECOG 1 status was in 18 (69%) patients (Table 2).

Patients in CBI + ICA stenosis group complained of headache (n = 13, 61%), dizziness (n = 11, 52%), memory loss (n = 16, 76%), fatigue (n = 11, 52%). Vestibulo-ataxic syndrome was detected in 19 patients (90%) and clinical manifestations of polyneuropathy in 6 patients (28%) (Table 3).

Patients in the CBI group also complained of headache (n = 29, 82%), dizziness (n = 20, 57%), memory loss (n = 26, 74%), fatigue (n = 22, 62%). Vestibulo-ataxic syndrome was detected in 30 patients (83%) and clinical manifestations of polyneuropathy in 8 patients (22%) (Table 3).

The serum biomarker levels in the study groups are presented in Table 4, Figure 1 and Figure 2. 

As can be seen from Table 4 (data are presented as median and ranges of values), in women after BC treatment level of NSE was 14.81 (12.75; 18.09), anti-NR-2 antibodies–0.45 (0.29; 0.76), ICAM-1–558.50 (510.50; 651.50), PECAM-1–104.15 (82.33; 128.75). In the group of healthy volunteers the level of NSE constituted 14.00 (11.65; 15.77), anti-NR-2 antibodies–1.28 (1.02; 1.57), ICAM-1–210.50 (196.50; 256.25), and PECAM-1–64.00 (49.75; 72.25). In the group of CBI + ICA stenosis level of NSE was 40.49 (21.42; 83.09), anti-NR-2 antibodies–0.41 (0.36; 0.75), ICAM-1–624.00 (562.00; 668.00), PECAM-1–96.80 (85.40; 113.80). In the group of CBI level of NSE was 31.92 (16.42; 51.17), anti-NR-2 antibodies–0.43 (0.32; 0.59), ICAM-1–574.00 (509.50; 610.00), PECAM-1–95.90 (78.65; 106.35).

The Kruskal–Wallis test for all groups is presented in Table 5.

As can be seen from Table 6, in all four groups the difference between the studied indicators turned out to be statistically significant (*p* < 0.001).

The Games–Howell test result for two groups according to a certain marker is presented in Table 6.

As can be seen from Table 6, in women after BC treatment, there was a significant increase in the level of ICAM-1(mean difference: −368.56, 95% CI −450.30 to −286.69, *p* < 0.001), PECAM-1 (mean difference: −47.75, 95% CI −68.73 to −26.77, *p* < 0.001), and a decrease in the level of anti-NR -2 antibodies (mean difference: 0.89, 95% CI 0.41 to 1.48, *p* < 0.001) compared to the group of healthy volunteers. The NSE concentration in blood serum in patients after BC treatment was higher than in healthy volunteers, but the differences were not statistically significant (mean difference: −3.53, 95% CI −13.23 to 6.18, *p* = 0.76).

In the group of CBI + ICA stenosis patients, an increase in serum levels of PECAM-1 (mean difference: −42.72, 95% CI −62.04 to −23.39, *p* < 0.001), ICAM-1 (mean difference: −402.18, 95% CI −461.08 to −243.27, *p* < 0.001), NSE (mean difference: −47.17, 95% CI −86.93 to −7.42, *p* = 0.02) molecules was detected, as well as a decrease in anti-NR-2 antibodies ((mean difference: 0.94, 95% CI 0.46 to 1.42, *p* < 0.001) levels compared to the group of healthy volunteers. In patients with CBI, an increase in serum levels of ICAM 1 (mean difference: −347.32, 95% CI −402.06 to −292.58, *p* < 0.001), NSE (mean difference: −40.11, 95% CI −73.84 to −6.39, *p* = 0.01), as well as a decrease in levels of anti-NR-2 antibodies (mean difference: 0.96, 95% CI 0.49 to 1.44, *p* < 0.001) compared to the group of healthy volunteers.

The intergroup comparison revealed comparable levels of ICAM-1 (mean difference: −33.58, 95% CI −58.10 to 125.26, *p* = 0.76), PECAM-1 (mean difference: −5.03, 95% CI −29.93 to 19.87, *p* = 0.95), as well as anti-NR-2 antibodies (mean difference: −0.05, 95% CI −0.26 to 0.16, *p* = 0.93) in patients after BC treatment and in patients with CBI + ICA stenosis. The NSE level in group CBI + ICA stenosis was significantly higher than in women after BC treatment (mean difference: −43.64, 95% CI 3.31 to -83.99, *p* = 0.03). Comparable levels of ICAM-1 were also found in patients after BC treatment and in the group of CBI (mean difference: −21.28, 95% CI −111.03 to 68.48, *p* = 0.92) and comparable levels of anti-NR-2 antibodies in these groups (mean difference: −0.07, 95% CI −0.27 to 0.12, *p* = 0.75). The level of PECAM-1 in patients after BC treatment) was also comparable to the group of CBI (mean difference: −13.68, 95% CI −35.51 to 8.15, *p* = 0.35). The level of NSE was lower in group of CBI (mean difference: 36.58, 95% CI 2.09 to 71.08, *p* = 0.03) than in women after BC treatment.

## 4. Discussion

In our study significantly increased levels of ICAM-1, PECAM-1 were found in patients following BC treatment. In the current study, the authors conducted a comprehensive analysis of the symptoms of damage to the central and peripheral nervous system in patients after BC treatment. Neurological and clinical examination revealed local and systemic treatment-related symptoms in patients in the long-term follow-up period. The group of patients in the study includes women who all have undergone relatively radical treatments, both of which are known to cause significant regional symptoms and does not include women who received less-aggressive treatment such as lesser surgeries (e.g., lumpectomies), no chemotherapy or no radiotherapy. Therefore, the obtained findings cannot be applied to women receiving milder forms of therapy. It should be noted that patients after BC treatment, have a similar high risk to develop damages in the CNS and endothelium, than elderly patients with atherosclerotic lesions. The increased level of soluble adhesion molecules in the blood of patients after BC treatment might indicate endothelial dysfunction, which could predict CNS vascular damage and an increased risk factor for the development of neurological manifestations. A disruption of the endothelium leads to chronic brain ischemia, which causes microstructural damage to the white matter of the brain and a decrease in synaptogenesis. Some authors have also considered subclinical hypoxia as the main pathogenetic factor for the development of cancer-related CNS damage [61], but its specific molecular mechanisms have not yet been defined [62]. It should also be noted that intercellular adhesion molecules are indicative for inflammation [63]. Williams et al. have recently discovered increased levels of pro-inflammatory molecules (in particular, tumor necrosis factor-alpha, monocyte chemotactic protein-1) in the blood of patients with BC after chemotherapy, and demonstrated a link to visual memory impairment [64], while other researchers have found significantly increased concentrations of interleukins during therapy [65]. The relationship of elevated levels of IL-6 and TNF-alpha with verbal memory impairment and a decrease in the volume of the left hippocampus in BC patients log-term after therapy was demonstrated by Kesler et al. [65]. Therefore, it is assumed that a chronic inflammation could cause CNS damage in this group of patients.

The experimental group does not represent a typical group of patients with cancer-related cognitive impairment (CRCI), as cognition was not specifically tested. 

The involvement of the central nervous system in the experimental group was indirectly confirmed by systemic complaints of patients and the presence of the vestibulo-atactic syndrome. The limitation of the study is also that the CBI group included men. However, gender differences in the levels of these biomarkers have not been reliably proven to date. Thus, gender differences should not affect the results of the study.

Considering that in the process of endothelium destruction there is an active release of proinflammatory cytokines, a promising area of research in the future may be the study of levels of interleukin-6, tumor necrosis factor-alpha in patients after breast cancer treatment and their comparison with the levels of intercellular adhesion molecules. In addition, determination of circulating endothelial progenitor cells can be used to verify endothelial damage, which has been proven in recent studies [66].

The NSE level in the group of patients after BC treatment exceeded the reference values and was significantly higher than in the control group, although statistical significance was not reached, most likely due to the insufficient sample size of the experimental group. The concentration of NSE in the groups of patients with CBI, both without hemodynamically significant ICA stenosis and with atherosclerotic ICA stenosis, significantly exceeded the control and reference values. NSE is usually used as a marker of immediate neural damage. The chemotherapeutic agents direct a neurotoxic effect on CNS as the possible trigger for cognitive impairment as discussed by many researchers [67]. It was previously assumed that only methotrexate and 5-fluoroacil can cross the BBB. However, recent studies have shown that almost all chemotherapeutic drugs at low concentrations can cross the BBB and cause a variety of pathological processes [68]. For instance, during PET scans low concentrations of radioactively-labeled cysplastine and paclitaxel (that are most commonly used for BC treatment) could be detected in the brain tissue [69,70]. Oxidative cell stress is a well-studied effect of chemotherapy, associated with an increase of free radicals and an inhibition of intracellular antioxidative mechanisms [71]. NSE, as a marker of oxidative neural damage, might indicate a direct neurotoxic effect of chemotherapeutic agents. Moreover, NSE, according to literature, may reflect the degree of pathological neuroglial activation. In addition to chemotherapy, radiation also has a negative effect on glial cells [72]. Therefore, elevated NSE levels could represent oxidative damage in neurons and microglial disruption upon treatment.

The levels of anti-NR-2 antibodies in the women after BC treatment and CBI groups were significantly lower compared to the healthy control group but did not differ significantly between patients after BC treatment and patients with CBI. It is known that glutamate-mediated excitotoxicity is the main pathological process that occurs in many types of acute and chronic processes that damage the brain and BBB. Peptides formed during the cleavage of NMDA receptors can enter the bloodstream and, at increased concentrations, can induce autoimmune reactions. Anti-NR2 antibodies levels decreased differently in women after BC treatment. An immunosuppression caused by antineoplastic treatment could lower the antibody generation. However, the duration of changes induced doubts into this theory. The decrease in anti-NR2 antibodies could be caused by an exhaustion of glutamate-mediated mechanisms in the CNS. It is known that NMDA receptors are the major CNS receptors for synaptic plasticity control [73], and play an important role in development, learning and memory [74]. It is possible that prolonged hypoxia and chronic damage of the CNS tissue could gradually decrease the number of these receptors, and thereby might indirectly influencing the antibody concentrations in the peripheral blood. In addition, it should also be noted that BBB permeability may play central role. Short-term effects of radiation treatment on BBB architecture and function are well studied and include the increase of permeability, and regulatory mechanisms disruption [75]. However long-term changes that appear in the remote period after the damage have scarcely been investigated. Thus, low levels of anti-NR2 antibodies might reflect the BBB disruption in the remote period after treatment.

The results of the study indicate that endothelial dysfunction may be among the leading mechanisms of brain damage in women after BC treatment. To confirm this assumption, further studies are necessary using alternative methods for studying endothelial cell function, the correlation of the results of imaging techniques with the levels of biomarkers, the clinical picture, and their observation in dynamics. Verification of endothelial cell dysfunction, neuronal damage, and BBB dysfunction will allow us to create a pathogenetically reasonable treatment schedule and design prevention strategies for patients with post-treatment symptoms.

## 5. Conclusions

Currently, a correct and timely diagnosis of neurological disorders in breast cancer survivors is not possible. Due to the various clinical manifestations of the CNS damage, which present as cerebrovascular, emotional, and cognitive disorders, there is a high medical need for improved diagnostic methods. In theory, the evaluation of the kinetics of biomarkers in liquid biopsies could provide a useful tool to determine the degree of neuronal damage following breast cancer treatment and for the assessment of brain damage. In this work, the authors identified and evaluated some markers of CNS lesions in female patients after treatment for breast cancer. Our results suggested that the leading mechanism of damage is endothelial dysfunction. Further studies are required to confirm this hypothesis including the analysis of highly selective biomarkers of endothelial damage, their correlation with clinical picture and imaging data, and instrumental evaluation of endothelial function. Clarification of pathogenetic pathways of CNS damage will help to develop reasonable strategies for prevention and treatment of complications after breast cancer treatment.

## Figures and Tables

**Figure 1 jcm-11-01215-f001:**
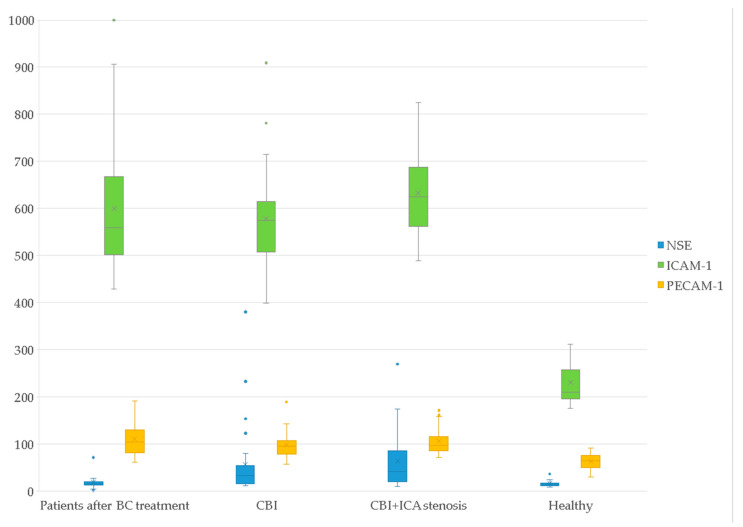
Serum levels of NSE, ICAM-1, and PECAM-1 molecules in the study groups (ng/mL).

**Figure 2 jcm-11-01215-f002:**
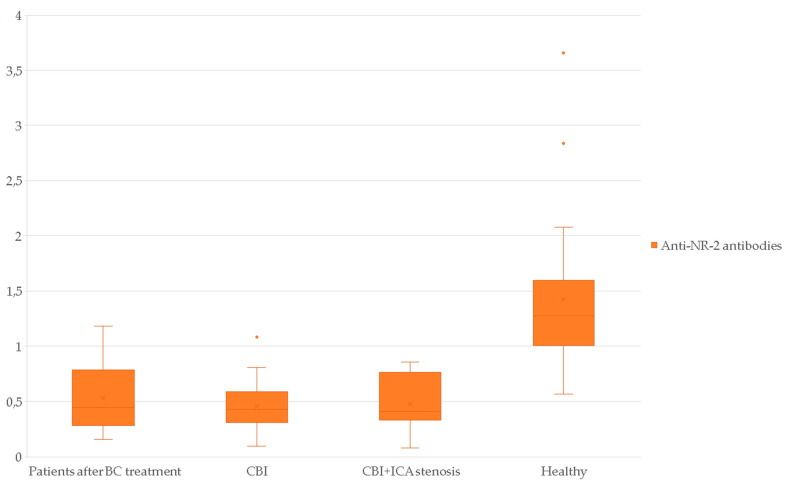
Serum Anti-NR-2 antibodies level in the study groups (ng/mL).

**Table 1 jcm-11-01215-t001:** Characteristics of the patients presented after breast cancer (BC) treatment.

Group Characteristics of Patients	Patients (N) n = 26	Patients (%)
Age (years)	47.0 ± 3.7	
Years since treatment	4.3 ± 2.1	
TNM stage		
I (T1N0M0)	3	12%
II A (T2N1M0)	6	23%
II B (T3N1M0)	7	7%
III A (T3N2M0)	5	19%
III B(T4N2M0)	5	19%
Treatment for breast cancer		
Complex treatment (surgical, radiotherapy, chemotherapy)	13	50%
Combination of surgical treatment and chemotherapy	8	30%
Combination of surgical treatment and radiotherapy	5	20%
Hormonal therapy (tamoxifen vs. growth hormone and luteinizing hormone (GH-LH) analogues)		
do not take the medicine	3	12%
take the medicine	16	62%
completed the course	7	26%

**Table 2 jcm-11-01215-t002:** Symptoms and complaints in patients after BC treatment.

Complaints and Symptoms	N	%
Headache	13	50%
Dizziness	15	57%
Memory degradation	9	34%
Fatigue	11	42%
Sleep disturbances	5	20%
Edema of the upper limb	11	42%
Decreased sensitivity of the upper limb	16	61%
Paresthesia or hypestesia	8	30%
Muscle weakness	18	69%
Limited movement in the shoulder	8	30%
Pain in the upper limb	13	50%
Pain in the upper arm	11	42%
Vestibulo-atactic syndrome	12	46%
Polyneuropathy	11	42%
ECOG 0	8	31%
ECOG 1	18	69%

**Table 3 jcm-11-01215-t003:** Complaints and symptoms of patients with chronic brain ischemia (CBI) and CBI + internal carotid artery (ICA) stenosis.

Complaints and Symptoms	CBI + ICA Stenosis (n = 21)		CBI (n = 35)	
	N	%	N	%
Headache	13	61%	29	82%
Dizziness	11	52%	20	57%
Memory degradation	16	76%	26	74%
Fatigue	11	52%	22	62%
Vestibulo-atactic syndrome	19	90%	30	83%
Polyneuropathy	6	28%	8	22%

**Table 4 jcm-11-01215-t004:** Serum biomarker levels in the study groups (n = 102).

		N	Mean Value	Standard Deviation	Minimum Value	Maximum Value	Percentiles		
							25	50th (median)	75th
Neuron-specific enolase (NSE)	Healthy	20	15.47	6.36	7.64	36.04	11.65	14.00	15.77
	CBI + ICA stenosis	21	62.64	64.87	9.73	268.90	21.42	40.49	83.09
	CBI	35	55.58	73.49	10.50	380.00	16.42	31.92	51.17
	Women after BC treatment	26	18.99	16.82	1.09	75.78	12.75	14.81	18.09
	Total	102	39.84	56.15	1.09	380.00	13.54	17.82	45.59
Antibodies to the NR-2 subunit of the N-methyl-D-aspartate (NMDA) receptor (Anti-NR-2 antibodies)	Healthy	20	1.42	0.74	0.57	3.66	1.02	1.28	1.57
	CBI + ICA stenosis	21	0.48	0.23	0.08	0.86	0.36	0.41	0.75
	CBI	35	0.46	0.22	0.10	1.08	0.32	0.43	0.59
	Women after BC treatment	26	0.53	0.32	0.16	1.18	0.29	0.45	0.76
	Total	102	0.67	0.54	0.08	3.66	0.34	0.49	0.84
Intercellular adhesion molecule type-1 (ICAM-1)	Healthy	20	230.25	44.79	176.00	311.00	196.50	210.50	256.25
	CBI + ICA stenosis	21	632.43	88.03	489.00	825.00	562.00	624.00	668.00
	CBI	35	577.57	106.47	398.00	909.00	509.50	574.00	610.00
	Women after BC treatment	26	598.85	144.75	429.00	1000.00	510.50	558.50	651.50
	Total	102	526.19	181.32	176.00	1000.00	451.50	550.00	625.25
Platelet adhesion molecule type-1 (PECAM-1)	Healthy	20	62.85	15.60	30.00	91.00	49.75	64.00	72.25
	CBI + ICA stenosis	21	105.57	28.47	70.50	171.50	85.40	96.80	113.80
	CBI	35	96.92	25.20	56.50	189.40	78.65	95.90	106.35
	Women after BC treatment	26	110.60	35.52	61.40	191.80	82.33	104.15	128.75
	Total	102	95.51	32.00	30.00	191.80	74.83	90.25	109.00

**Table 5 jcm-11-01215-t005:** Kruskal–Wallis test for all studied groups.

	NSE	Anti-NR-2 Antibodies	ICAM-1	PECAM-1
H Kruskal–Wallis	26.919	39.359	51.505	35.570
Df	3	3	3	3
Asymp. Sig. (*p*)	<0.001	<0.001	<0.001	<0.001

**Table 6 jcm-11-01215-t006:** Games–Howell test result for two groups according to a certain marker.

Dependent Variable	(I) Criterion	(J) Criterion	Mean Difference (I–J)	Std. Error	Sig.	95% Confidence Interval (CI)	
						Lower Bound	Upper Bound
NSE	Healthy	Women after BC treatment	−3.53	3.59	0.76	−13.23	6.18
		CBI + ICA stenosis	−47.17 *	14.23	0.02	−86.93	−7.42
		CBI	−40.11 *	12.50	0.01	−73.84	−6.39
	CBI + ICA stenosis	Women after BC treatment	43.64 *	14.54	0.03	3.31	83.99
		CBI	7.06	18.83	0.98	−43.12	57.24
	CBI	Women after BC treatment	36.58 *	12.85	0.03	2.09	71.08
Anti-NR-2 antibodies	Healthy	Women after BC treatment	0.89 *	0.18	<0.001	0.41	1.38
		CBI + ICA stenosis	0.94 *	0.17	<0.001	0.46	1.42
		CBI	0.96 *	0.17	<0.001	0.49	1.44
	CBI + ICA stenosis	Women after BC treatment	−0.05	0.08	0.93	−0.26	0.16
		CBI	0.02	0.06	0.98	−0.14	0.19
	CBI	Women after BC treatment	−0.07	0.07	0.75	−0.27	0.12
ICAM-1	Healthy	Women after BC treatment	−368.56 *	30.10	<0.001	−450.30	−286.89
		CBI + ICA stenosis	−402.18 *	21.66	<0.001	−461.08	−343.27
		CBI	−347.32 *	20.60	<0.001	−402.06	−292.58
	CBI + ICA stenosis	Women after BC treatment	33.58	34.28	0.76	−58.10	125.26
		CBI	54.86	26.32	0.17	−15.17	124.89
	CBI	Women after BC treatment	−21.28	33.61	0.92	−111.03	68.48
PECAM-1	Healthy	Women after BC treatment	−47.75 *	7.79	<0.001	−68.73	−26.77
		CBI + ICA stenosis	−42.72 *	7.12	<0.001	−62.04	−23.39
		CBI	−34.07 *	5.51	<0.001	−48.68	−19.46
	CBI + ICA stenosis	Women after BC treatment	−5.03	9.33	0.95	−29.93	19.87
		CBI	8.65	7.53	0.66	−11.58	28.88
	CBI	Women after BC treatment	−13.68	8.17	0.35	−35.51	8.15

* Differences between the groups are significant at *p* < 0.05.

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
