# Peer review of "Potential Molecular Biomarkers of Central Nervous System Damage in Breast Cancer Survivors"

_jcm, 2022, doi:10.3390/jcm11051215_

Round 1

Reviewer 1 Report

The authors resubmitted the manuscript. It is a revised version.

The authors did not provide more precise data on the statistical package used (city, etc.). I suggest you have a look at the other approved manuscripts.

Table 3: There are no relevant statistical analyzes performed.

The first tables should also include% (not just n).

Author Response

On behalf of all co-authors I would like to thank the reviewer for the provided comments. We have revised the manuscript accordingly.

Comment (1): The authors did not provide more precise data on the statistical package used (city, etc.). I suggest you have a look at the other approved manuscripts.

Answer (1): This was provided.

Comment (2): Table 3: There are no relevant statistical analyzes performed.

Answer (2): We have corrected the table.

Comment (3): The first tables should also include% (not just n).

Answer (3): We have corrected the tables accordingly.

Reviewer 2 Report

This is a study with clinical relevance, because it provides another alternative for objective metrics (the biomarkers in question) to assess post-therapy sequelae in breast cancer patients. It is a simple, straightforward study that looks into breast cancer survivors which have undergone relatively radical treatments and whom present with lingering neuro-vascular symptomatology. The changes observed in the biomarkers seem to correlate with the symptomatology presented and upon further validation, may eventually be used in the clinic. Event though the implications of the data are clear for future clinical use, the authors need to be more focused in their conclusions, as they have cast too wide a net, making inferences that are not supported by the data.

1) English and writing.

The major weakness of the paper is in the English and the writing. This manuscript needs to be edited because the errors take away from the message and from the data. There are writing errors that denote some carelessness in the details. For example in line 297: “The anamnesis included (the period after the operation, the type of operation, the course of chemotherapy, the course of radiation therapy, the presence of relapses, and the use of Tamoxifen).”….what is the point of these parentheses? These types of errors that violate basic rules of writing can and should be remedied.

There are frequent inconsistencies in the terminology and abbreviations used that introduce confusion in the reading and understanding of the study. For instance, one of the experimental groups is CBI+ICA. This group is referred to as ICA in some occasions, causing confusion. It should consistently be called CBI+ICA, even in the figures.

Another notable mistake found in figure 1, is the term PMPS. This abbreviation seems to refer to patients who received BC treatment. This term however is not mentioned or used in any other section of the article and therefore, must be clarified or renamed.

Another example is the use of the terms hypesthesia and hypoesthesia. They both have the same meaning but only one should be used for consistency and clarity.

In the Experimental Design, in line 234, group 2 is described as patients with chronic cerebral ischemia (CBI) with asymptomatic internal carotid artery (ICA) stenosis and group 3 is presented as patients with CBI without asymptomatic ICA stenosis. Group 3 could be interpreted as either patients with CBI+ICA (that is symptomatic), or patients with CBI without ICA. This needs to be clarified.

There are terms that are used incorrectly, as when the word ‘subjective’ is used in line 300: “Subjective examination of the еxperimental group (women after BC treatment) included: neurological examination, measurement of the volume of the upper extremities, joints movements.” In this case, the word that should be used is objective, not subjective.

In Inclusion Criteria, once again, consistency in terminology is essential for clarity. In line 242, the authors refer to a treatment as a combination of surgical treatment and systemic therapy. Right after this, they mention complex treatment as a combination of surgery, radiation therapy, and chemotherapy. Thus, it seems that systemic therapy and chemotherapy are the same. If so, only one of these terms should be used throughout the manuscript.

2) Scientific soundness.

The non-inclusion of men in the BC treatment group is understandable. However, the other groups used for comparison, should have included women only, since biomarker levels can vary between sexes. I can understand that the authors probably prioritized higher sample numbers, but this denotes lack of rigorous scientific practice. It does appear that in the end, the results do not look as though they were skewed due to the issue of sex and that the differences between biomarkers were significant regardless of sex. However, it is imperative the authors discuss this weakness.

In the Clinical and Neurological Assessment section, it seems the authors performed a careful and complete clinical examination. The question is: what was the point of this? Was it to confirm that the clinical history was sound and consistent with the treatment received? And if so, what was the intention? To select patients who still (years after their treatments ended) presented with relevant symptoms? Were asymptomatic patients disqualified from the study? Please clarify.

One very important benefit of biomarkers is the possibility of these being capable of reflecting the degree of clinical compromise. Did the authors consider correlating the serum levels of the biomarkers with the number, character and severity of the symptoms in the group of patients with BC treatment? If not, why not? Was the sample too small? Did they try assessing these associations but did not find any correlation? This needs to be addressed and explained.

3) Discussion.

The results seem to be incontrovertible. However, some of the conclusions are extrapolated to other conditions, without the data supporting this.

The most important issue with the manuscript is the definition of the experimental group. The group of patients in the study includes women who all have undergone relatively radical treatments, which include surgery and radiotherapy, both of which are known to cause significant regional symptoms such as hypoesthesia, muscle weakness and edema, as described in the report. Thus, this is a very specific type of treatment group that shares a certain level of radical treatment, all of which are symptomatic, most of them presenting with regional compromise. However, in some instances, the authors seem to try to conflate their findings to apply them to all women who have received BC treatment. Unfortunately, their findings cannot be extrapolated to other treatment groups. It is necessary to clarify that the group studied does not include women who received less aggressive treatments such as lesser surgeries (lumpectomies, for example), no chemotherapy or no radiotherapy and that therefore, these data cannot be applied to women receiving milder forms of therapy. These biomarkers reflect a constellation of symptoms, most of which are regional, peripheral and secondary to radical treatment; not other groups.

The experimental group does not represent a typical group of patients with cancer-related cognitive impairment (CRCI), as most of the symptoms were peripheral and cognition was not specifically tested. This does not take away from the findings, as the serum biomarkers are still an objective way of measuring the degree of compromise of this specific group with these specific symptoms. But the manuscript must be clear about this and not make inferences regarding CRCI, when the data does not provide information in that respect. CRCI has been found to be associated with CNS neuronal damage, white matter compromise and glial involvement. But none of these aspects were tested and once again, most of the symptoms in the study were peripheral, not CNS-related. Therefore, the discussion must be more focused and not stray into making inferences that are not supported by the data.

In the interpretation of the results, it is important to consider the significance of the biomarkers, not only in terms of their levels, but also in relation to their probable origins. For instance, several of these biomarkers are known to be elevated with many pathologies that are not centered in the CNS. Thus, these biomarkers can well reflect the neurological or vascular compromise of regions in the periphery, not the CNS.

The elevated biomarkers seem to reflect a chronic condition that might reflect CNS damage, but might also reflect peripheral neurovascular involvement. Also, the data can imply involvement of neuro-inflammation (whether CNS-related or peripheral) and/or endothelial/vascular involvement (whether CNS-related or peripheral). Thus, the statement that the data reflect endothelial damage primarily, should not be made.

NSE is mostly a biomarker of acute conditions due to its half-life of ~24 h. It was the biomarker with the least change, which can also support the theory that these biomarkers perhaps being associated mostly with chronic damage.

Author Response

On behalf of all co-authors I would like to thank the reviewer for provided comments. We have revised our manuscript accordingly.

Comment (1): The major weakness of the paper is in the English and the writing. This manuscript needs to be edited because the errors take away from the message and from the data. There are writing errors that denote some carelessness in the details. For example in line 297: “The anamnesis included (the period after the operation, the type of operation, the course of chemotherapy, the course of radiation therapy, the presence of relapses, and the use of Tamoxifen).”….what is the point of these parentheses? These types of errors that violate basic rules of writing can and should be remedied.

Answer (1): This we corrected.

Comment (2): There are frequent inconsistencies in the terminology and abbreviations used that introduce confusion in the reading and understanding of the study. For instance, one of the experimental groups is CBI+ICA. This group is referred to as ICA in some occasions, causing confusion. It should consistently be called CBI+ICA, even in the figures.

Answer (2): This was corrected.

Comment (3): Another notable mistake found in figure 1, is the term PMPS. This abbreviation seems to refer to patients who received BC treatment. This term however is not mentioned or used in any other section of the article and therefore, must be clarified or renamed.

Answer (3): This was corrected.

Comment (4): Another example is the use of the terms hypesthesia and hypoesthesia. They both have the same meaning but only one should be used for consistency and clarity.

Answer (4): This was corrected.

Comment (5): In the Experimental Design, in line 234, group 2 is described as patients with chronic cerebral ischemia (CBI) with asymptomatic internal carotid artery (ICA) stenosis and group 3 is presented as patients with CBI without asymptomatic ICA stenosis. Group 3 could be interpreted as either patients with CBI+ICA (that is symptomatic), or patients with CBI without ICA. This needs to be clarified.

Answer (5): This we clarified.

Comment (6): There are terms that are used incorrectly, as when the word ‘subjective’ is used in line 300: “Subjective examination of the еxperimental group (women after BC treatment) included: neurological examination, measurement of the volume of the upper extremities, joints movements.” In this case, the word that should be used is objective, not subjective.

Answer (6): This was corrected.

Comment (7): In Inclusion Criteria, once again, consistency in terminology is essential for clarity. In line 242, the authors refer to a treatment as a combination of surgical treatment and systemic therapy. Right after this, they mention complex treatment as a combination of surgery, radiation therapy, and chemotherapy. Thus, it seems that systemic therapy and chemotherapy are the same. If so, only one of these terms should be used throughout the manuscript.

Answer (7): We clarified this in the Materials and Methods sections as follows: "The study included women aged 25 to 50 after BC treatments including a combination of surgical treatment (unilateral or bilateral Peutie mastectomy) and radiation therapy, a combination of surgical treatment and chemotherapy, or a complex treatment (combination of surgery, radiation therapy, and chemotherapy)".

Comment (8): The non-inclusion of men in the BC treatment group is understandable. However, the other groups used for comparison, should have included women only, since biomarker levels can vary between sexes. I can understand that the authors probably prioritized higher sample numbers, but this denotes lack of rigorous scientific practice. It does appear that in the end, the results do not look as though they were skewed due to the issue of sex and that the differences between biomarkers were significant regardless of sex. However, it is imperative the authors discuss this weakness.

Answer (8): We totally agree with a reviewer and have included this limitation of the study into the discussion section.

Comment (9): In the Clinical and Neurological Assessment section, it seems the authors performed a careful and complete clinical examination. The question is: what was the point of this? Was it to confirm that the clinical history was sound and consistent with the treatment received? And if so, what was the intention? To select patients who still (years after their treatments ended) presented with relevant symptoms? Were asymptomatic patients disqualified from the study? Please clarify.

Answer (9): We intended to include only the patients with symptoms. Patients without post-treatment symptoms were not included in the study. We have added this notion into the Materials and methods section.

Comment (10): One very important benefit of biomarkers is the possibility of these being capable of reflecting the degree of clinical compromise. Did the authors consider correlating the serum levels of the biomarkers with the number, character and severity of the symptoms in the group of patients with BC treatment? If not, why not? Was the sample too small? Did they try assessing these associations but did not find any correlation? This needs to be addressed and explained.

Answer (10): We intended to assess the correlation between the severity of the clinical manifestations and the serum levels of the markers. However, the number of patients for further subgroup analysis was quite small. We plan in our future trials to expand the groups of enrolled patients to perform this analysis.

Comment (11): The most important issue with the manuscript is the definition of the experimental group. The group of patients in the study includes women who all have undergone relatively radical treatments, which include surgery and radiotherapy, both of which are known to cause significant regional symptoms such as hypoesthesia, muscle weakness and edema, as described in the report. Thus, this is a very specific type of treatment group that shares a certain level of radical treatment, all of which are symptomatic, most of them presenting with regional compromise. However, in some instances, the authors seem to try to conflate their findings to apply them to all women who have received BC treatment. Unfortunately, their findings cannot be extrapolated to other treatment groups. It is necessary to clarify that the group studied does not include women who received less aggressive treatments such as lesser surgeries (lumpectomies, for example), no chemotherapy or no radiotherapy and that therefore, these data cannot be applied to women receiving milder forms of therapy. These biomarkers reflect a constellation of symptoms, most of which are regional, peripheral and secondary to radical treatment; not other groups.

Answer (11): We agree with a reviewer and have added this notion into the Discussion section as follows: In the current study, the authors conducted a comprehensive analysis of the symptoms of damage to the central and peripheral nervous system in patients after BC treatment. Neurological and clinical examination revealed local and systemic treatment-related symptoms in patients in the long-term follow-up period. The group of patients in the study includes women who all have undergone relatively radical treatments, both of which are known to cause significant regional symptoms and does not include women who received less aggressive treatments such as lesser surgeries (e.g., lumpectomies), no chemotherapy or no radiotherapy. Therefore, the obtained findings cannot be applied to women receiving milder forms of therapy. 

Comment (12): The experimental group does not represent a typical group of patients with cancer-related cognitive impairment (CRCI), as most of the symptoms were peripheral and cognition was not specifically tested. This does not take away from the findings, as the serum biomarkers are still an objective way of measuring the degree of compromise of this specific group with these specific symptoms. But the manuscript must be clear about this and not make inferences regarding CRCI, when the data does not provide information in that respect. CRCI has been found to be associated with CNS neuronal damage, white matter compromise and glial involvement. But none of these aspects were tested and once again, most of the symptoms in the study were peripheral, not CNS-related. Therefore, the discussion must be more focused and not stray into making inferences that are not supported by the data.

Answer (12): We have added this notion concerning the CRCI into the discussion section.

Comment (13): In the interpretation of the results, it is important to consider the significance of the biomarkers, not only in terms of their levels, but also in relation to their probable origins. For instance, several of these biomarkers are known to be elevated with many pathologies that are not centered in the CNS. Thus, these biomarkers can well reflect the neurological or vascular compromise of regions in the periphery, not the CNS.

The elevated biomarkers seem to reflect a chronic condition that might reflect CNS damage, but might also reflect peripheral neurovascular involvement. Also, the data can imply involvement of neuro-inflammation (whether CNS-related or peripheral) and/or endothelial/vascular involvement (whether CNS-related or peripheral). Thus, the statement that the data reflect endothelial damage primarily, should not be made.

Answer (13): we agree we the reviewer that one cannot completely reject the assumption that peripheral symptoms could also affect the levels of the biomarkers studied. In this study, a small sample did not allow for a correlation between the severity of systemic and peripheral symptoms and biomarker levels. This does not allow us to make an unambiguous conclusion that the damage of the central nervous system is the main reason for the increase in their level. Further studies are needed, including additional tests and instrumental methods to assess the central nervous system in patients after treatment.

Comment (14): NSE is mostly a biomarker of acute conditions due to its half-life of ~24 h. It was the biomarker with the least change, which can also support the theory that these biomarkers perhaps being associated mostly with chronic damage.

Answer (14): We would like to thank the reviewer for this notion. In future trial we were going to additionally assess the levels of NSE and the changes of the functional network connectivity of the patients employing the fMRI.

This manuscript is a resubmission of an earlier submission. The following is a list of the peer review reports and author responses from that submission.

Round 1

Reviewer 1 Report

The work entitled "Potential molecular biomarkers of central nervous system damage in breast cancer survivors" studies endothelial dysfunction caused by cancer/treatment. It is an interesting work that addresses a clinical fact that has previously been demonstrated and studied. However, in this work, the authors associate cognitive impairment with previous endothelial disruption. I would like to highlight certain points:

1. There are various grammatical errors throughout the text that should be corrected.
2. For better visualization, the data in table 3 should also be converted into graphs showing the variation and statistics.
3. In any endothelial disruption process, stimulation of the inflammatory response occurs, such as an increase in pro-inflammatory cytokines such as IL-6 or TNF-alpha that should be studied and correlated with the adhesion molecules. A new experiment should be designed and presented.
4. Recent studies have shown the importance of circulating EPCs in endothelial regeneration. Have the authors studied the% of EPCs in the blood of the different groups?

Author Response

We would like to thank the reviewer for the provided comments. We have extensively revised the manuscript addressing all the provided critical points.

Comment (1): There are various grammatical errors throughout the text that should be corrected.

Answer (1): We have corrected the grammatical errors throughout the manuscript and rewritten certain paragraphs.

Comment (2). For better visualization, the data in table 3 should also be converted into graphs showing the variation and statistics.

Answer (2): As suggested we have incorporated two new figures for better visualization of the presented data. Furthermore, we have provided new Tables 5 and 6 where we have employed the ANOVA statistical analysis.

Comment (3): In any endothelial disruption process, stimulation of the inflammatory response occurs, such as an increase in pro-inflammatory cytokines such as IL-6 or TNF-alpha that should be studied and correlated with the adhesion molecules. A new experiment should be designed and presented.

Answer (3): We totally agree with a reviewer about the importance of IL-6 and TNFa in the endothelial dysfunction. We have added this notion into the Discussion section as follows:  "Considering that in the process of endothelium destruction there is an active release of proinflammatory cytokines, a promising area of research in the future may be the study of levels of interleukin-6, tumor necrosis factor-alpha in patients after breast cancer treatment and their comparison with the levels of intercellular adhesion molecules. In addition, determination of circulating endothelial progenitor cells can be used to verify endothelial damage, which has been proven in recent studies."

Comment (4): Recent studies have shown the importance of circulating EPCs in endothelial regeneration. Have the authors studied the% of EPCs in the blood of the different groups?

Answer (4): We totally agree that circulating EPCs are of high importance for endothelial regeneration. However, in the current study we did not evaluate the circulating EPCs. In future trials we will incorporate into the study protocol this parameter. We also have noted the importance of EPCs in the Discussion section.

Reviewer 2 Report

Summary

In general, this is a very interesting article and relevant in the field of study of breast cancer research for early detection of CNS dysfunction in patients.

Title and authorship

Comments: Attractive title

Abstract

Comments: Aim and purpose clearly stated however there is some confusion with the interpretation of results.

       - Abbreviations (line 30). A review of punctuation in sentences is required.

 - In list of abbreviations missing PMPS, BCNU

Introduction

Comments: Very interesting read, and citations were well chosen to support and expand ideas.

  • The particular reason to the problematic were not properly described before presenting the aims of the study. There could be more expansion on breast cancer in particular treatments available before stating the limitations which make it seems rushed.
  • In line 73 and 81 the use of the word ‘could’ is confusing as it suggests there were no preliminary research done.
  • The reason why these particular biomarkers were chosen in relation to damage of the CNS needs additional support in the literature.
  • It is unclear why patients with chronic ischemic brain disorder are used as a compare group to breast cancer.

Materials and methods

Comments: Each test should be stated clearly and then defined in particular sensory and motor tests performed on patients.

  • Is there are reason why there are no males in groups 1 and 4? Maybe it could be stated as to the reasons for this? Is the study sex dependent?
  • The potential or lack risks to patients during the study is unclear.
  • ECOG status is undefined (line 211).

Results

Comments:  There seem to be a lack of information as to what was described in the previous section. Difficulty in following explanation after results were presented in the tables.

  • As the paper is focusing on potential biomarkers of CNS damage, there is merit in sharing the results of the sensory perception, motor function and coordinated movements tests results of the different groups as these can be further proof to support CNS dysfunction.
  • Table 1, if there are null values in relation to characteristics it may be best to eliminate them from table.
  • Table 3, last row there are commas in place of dots for values. The numbers need to be consistent to significant values. In general, it is a bit difficult to glance at, very encumbered.
  • Keep abbreviations when describing the groups all through results it makes it easier to follow the work (examples: line 350, 356).

Discussion

Comments: Explained results and supported with other bodies with work. Provides explanation for the biomarkers changes in the different groups. However, there is a lack of the possible next steps to expand study.

  • Is there any research on how of anti-NR-2 antibodies change in men after BC?
  • Lack of annotation for the very last figure, inappropriately placed in test and lacks information. What does it represent?

Conclusion

Comments: Proposes potential next step but not well explained and did not conclude on the aims of this study.

Author Response

We would like to thank the reviewer for the provided comments. We have extensively revised the manuscript according to the provided critical points.

Comment (1): 

Summary

In general, this is a very interesting article and relevant in the field of study of breast cancer research for early detection of CNS dysfunction in patients.

Title and authorship

Comments: Attractive title

Abstract

Comments: Aim and purpose clearly stated however there is some confusion with the interpretation of results.

       - Abbreviations (line 30). A review of punctuation in sentences is required.

 - In list of abbreviations missing PMPS, BCNU

Answer (1): We have corrected the abstract including the punctuation. Additionally, we added the missing abbreviations.

Comment (2):

Introduction

Comments: Very interesting read, and citations were well chosen to support and expand ideas.

  • The particular reason to the problematic were not properly described before presenting the aims of the study. There could be more expansion on breast cancer in particular treatments available before stating the limitations which make it seems rushed.
  • In line 73 and 81 the use of the word ‘could’ is confusing as it suggests there were no preliminary research done.
  • The reason why these particular biomarkers were chosen in relation to damage of the CNS needs additional support in the literature.
  • It is unclear why patients with chronic ischemic brain disorder are used as a compare group to breast cancer.

Answer (2): We have extensively revised the Introduction section adding new references to support the manuscript.

Comment (3): 

Materials and methods

Comments: Each test should be stated clearly and then defined in particular sensory and motor tests performed on patients.

  • Is there are reason why there are no males in groups 1 and 4? Maybe it could be stated as to the reasons for this? Is the study sex dependent?
  • The potential or lack risks to patients during the study is unclear.
  • ECOG status is undefined (line 211).

Answer (3): We have defined the ECOG status. ECOG 0 status was in 8 (30%) patients and ECOG 1 status was in 18 (69%) patients (Table 2).

In this research, the men were excluded from the study due to the low prevalence of BC diagnosis - less than 1% of all BC cases [reference 57], which can distort the statistical processing of data including the differences of both sexes in the expression of intercellular adhesion molecules and the production of proinflammatory cytokines. The researchers also found no information about the biomarker concentration in men with BC.

Comment (4): 

Results

Comments:  There seem to be a lack of information as to what was described in the previous section. Difficulty in following explanation after results were presented in the tables.

  • As the paper is focusing on potential biomarkers of CNS damage, there is merit in sharing the results of the sensory perception, motor function and coordinated movements tests results of the different groups as these can be further proof to support CNS dysfunction.
  • Table 1, if there are null values in relation to characteristics it may be best to eliminate them from table.
  • Table 3, last row there are commas in place of dots for values. The numbers need to be consistent to significant values. In general, it is a bit difficult to glance at, very encumbered.
  • Keep abbreviations when describing the groups all through results it makes it easier to follow the work (examples: line 350, 356).

Answer (4): We have restructured the Results section adding two new figures for better visualization of the obtained data. Additionally, we have included Tables 5 and 6 where we have included the ANOVA analysis of the studied patients groups.

Comment (5): 

Discussion

Comments: Explained results and supported with other bodies with work. Provides explanation for the biomarkers changes in the different groups. However, there is a lack of the possible next steps to expand study.

Answer (5): We have rewritten the Discussion section. Thus, we have incorporated the notion that other cytokines (eg IL-6, TNFa) and circulating endothelial progenitor cells (EPCs) could be incorporated into further study protocols as follows: "Considering that in the process of endothelium destruction there is an active release of proinflammatory cytokines, a promising area of research in the future may be the study of levels of interleukin-6, tumor necrosis factor-alpha in patients after breast cancer treatment and their comparison with the levels of intercellular adhesion molecules. In addition, determination of circulating endothelial progenitor cells can be used to verify endothelial damage, which has been proven in recent studies [67]."

Comment (6): 

  • Is there any research on how of anti-NR-2 antibodies change in men after BC?

Answer (6): We found no information about the biomarker anti-NR-2 antibodies concentration in men with BC.

Comment (7): 

  • Lack of annotation for the very last figure, inappropriately placed in test and lacks information. What does it represent?

Answer (7): For clarity we have deleted the figure from the Discussion section.

Comment (8): 

Conclusion

Comments: Proposes potential next step but not well explained and did not conclude on the aims of this study.

Answer (8): We have rewritten the Conclusion section accordingly.

Reviewer 3 Report

The abstract does not reflect the results obtained.

The authors described too briefly potential biomarkers (their connection with the described disease, detailed description of their role, previous research on this topic, etc.).

The purpose of the research has not been thoroughly described.

No approval number from the Bioethics Committee.

“(M±m)” - Incorrect entry.

Statistical analyses and results

This is the main drawback of the article and the main reason for rejection.

Table 2 - no statistical test results

“When comparing the two groups, the nonparametric Mann-Whitney and Wil- 303 coxon criteria were employed.”

There are more than 2 groups here! The statistical tests used are incorrect. Statistical test results are not recorded according to scientific standards. The calculated effect size as well as the power of the statistical tests used are missing. It is not known where and what statistical test was used. To sum up, the results do not reflect reality and are subject to a large error.

The conclusions are not supported by a reliable analysis.

The discussion is not divided into logical sections.

References are not written uniformly.

“Spearman's correlation analysis was used” - The article does not contain reliable results of these analyzes and characteristic figures.

Author Response

We would like to thank the Reviewer for the provided comments. We have extensively revised the manuscript according to the provided critical points. Additionally, we have employed statistical analysis as was proposed by the reviewer.

Comment (1): The abstract does not reflect the results obtained.

Answer (1): We have rewritten the abstract.

Comment (2): 

The authors described too briefly potential biomarkers (their connection with the described disease, detailed description of their role, previous research on this topic, etc.).

The purpose of the research has not been thoroughly described.

Answer (2): We have rewritten the Introduction section accordingly.

Comment (3): No approval number from the Bioethics Committee.

Answer (3): We have provided the number of the study protocol.

Comment (4): 

(M±m)” - Incorrect entry.

Statistical analyses and results

This is the main drawback of the article and the main reason for rejection.

Table 2 - no statistical test results

“When comparing the two groups, the nonparametric Mann-Whitney and Wil- 303 coxon criteria were employed.”

There are more than 2 groups here! The statistical tests used are incorrect. Statistical test results are not recorded according to scientific standards. The calculated effect size as well as the power of the statistical tests used are missing. It is not known where and what statistical test was used. To sum up, the results do not reflect reality and are subject to a large error.

Answer (4): We have reanalyzed the obtained data as was suggested by the Reviewer. Statistical processing of the obtained data was carried out using the IBM SPSS Statistics 28.0.1.0 program. When comparing the four groups, the ANOVA were employed. The results are presented as statistical significance. The Statistical power is 42,04%. The probability of type I error in an experiment of four comparisons is 26.49%. The probability of type I error in an experiment from pairwise comparisons is 5%. The differences were considered significant at p<0.05.

Additionally, we have added two new Figures 1 and 2, new Tables 5 and 6 for the comparison of the studied groups.

Comment (5): The conclusions are not supported by a reliable analysis.

Answer (5): We have rewritten the Conclusion section accordingly.

Comment (6): The discussion is not divided into logical sections.

Answer (6): We have corrected the Discussion section.

Comment (7): References are not written uniformly.

Answer (7): This was corrected.

Comment (8): “Spearman's correlation analysis was used” - The article does not contain reliable results of these analyzes and characteristic figures.

Answer (8): We have reanalyzed the provided data. Additionally, we added new Figures 1 and 2, Tables 5 and 6.

Round 2

Reviewer 1 Report

Although the work has been substantially improved, the presentation of the data should be improved for a better understanding.

Author Response

We would like to thank the reviewer for the provided comment. We have substantially revised the manuscript for a better understanding. Thus, we have modified the abstract providing the exact data concerning the intergroup analysis. Additionally, we have revised our statistical analysis presenting corrected Tables 4, 5, and 6.

Reviewer 3 Report

The article has not been corrected in terms of analyzes. As a biostatistics expert, I say that the statistical analysis is incorrectly carried out (and thus the discussion on the results and conclusions).

The results of statistical significance are missing in the abstract. Example: “were significantly higher” - that's definitely not enough described.

“When comparing the four groups, the ANOVA were employed.” - This is a mistake, the Kruskal-Wallis test should be used here (Small groups, unequal etc.), as well as the correct post-hoc test. Test results should be recorded according to scientific standards. Significance level (only!) - it is not enough.

The size of the effect is not described by specific coefficients, i.e. specific to specific statistical tests.

Spearman’s correlation - Please indicate the results of this analysis in the article.

The discussion and conclusions must be based on a fresh analysis.

Author Response

We would like to thank the reviewer for the provided comments. We have substantially revised the manuscript adding new statistical analysis. The answers to the raised comments are provided below.

Comment (1): The article has not been corrected in terms of analyzes. As a biostatistics expert, I say that the statistical analysis is incorrectly carried out (and thus the discussion on the results and conclusions).

The results of statistical significance are missing in the abstract. Example: “were significantly higher” - that's definitely not enough described.

Answer (1): We have revised our statistical analysis accordingly. Additionally, we have modified the abstract.

Comment (2): “When comparing the four groups, the ANOVA were employed.” - This is a mistake, the Kruskal-Wallis test should be used here (Small groups, unequal etc.), as well as the correct post-hoc test. Test results should be recorded according to scientific standards. Significance level (only!) - it is not enough.

Answer (2): We have employed Kruskal-Wallis test for the revision of our manuscript.

Comment (3):  Spearman’s correlation - Please indicate the results of this analysis in the article.

Answer (3): The notion of the Spearman's correlation in the manuscript was included by mistake. We did not perform this analysis so the sentence was deleted from the revised version.

Comment (4): The discussion and conclusions must be based on a fresh analysis.

Answer (4): We have reevaluated the provided data thus employing Kruskal-Wallis test. Additionally, in the revised version of the manuscript we have included revised tables 4, 5, and 6.